# Factors That Limit the Adoption of Biofloc Technology in Aquaculture Production in Mexico

**Erick Arturo Betanzo-Torres** [1,2], **María de los Ángeles Piñar-Álvarez** [1,*],
**Luis Carlos Sandoval-Herazo** [2,3], **Antonio Molina-Navarro** [4], **Isidro Rodríguez-Montoro** [2] and
**Raymundo Humberto González-Moreno** [5]

1   El Colegio de Veracruz, Carrillo Puerto No. 26, Xalapa C.P. 91000, Veracruz, Mexico;
    erickbetanzo@hotmail.com
2   Division of Postgraduate Studies and Research, Tecnológico Nacional de México/Instituto Tecnológico
    Superior de Misantla, Misantla C.P. 93821, Veracruz, Mexico; lcsandovalh@gmail.com (L.C.S.-H.);
    irodriguezm@misantla.tecnm.mx (I.R.-M.)
3   Division of Environmental Engineering, Tecnológico Nacional de México/Instituto Tecnológico Superior de
    Misantla, Misantla C.P. 93821, Veracruz, Mexico
4   Engineering School, Construction and Housing, Universidad Veracruzana Bv. Adolfo Ruíz Cortines 455,
    Costa Verde, Boca del Rio C.P. 94294, Veracruz, Mexico; molher67@hotmail.com
5   Division of Civil Engineering, Tecnológico Nacional de Méexico/Instituto Tecnológico Superior de Misantla,
    Misantla C.P. 93821, Veracruz, Mexico; hrgonzalezm@misantla.tecnm.mx
*   Correspondence: angelespinaralvarez@gmail.com; Tel.: +52-2281059115

**Abstract:** Aquaculture uses large volumes of water, which is generally discharged without treatment, possibly causing scarcity and contamination. A sustainable aquaculture option is biofloc technology (BFT), which recycles food residues and toxic organic and inorganic compounds from the system through microorganisms, avoiding excessive use of water and serving as natural food for cultured aquatic organisms. The aim of this study was to identify the main factors that limit a Mexican aquaculture producer from adopting biofloc technology in their aquaculture production units (APUs). Strengths and weaknesses were methodologically analyzed through 248 questionnaires, applied to fish farmers in 16 states of the country with a mixed approach (quantitative and qualitative). Findings reveal that the main obstacles in the use of BFT are due to the following: low academic level, limited administrative capacity, scarce technological equipment in facilities, diversified productive activity, and obsolete regulations. Other factors that promote the adoption of BTFs for aquaculturists are production experience, favorable weather conditions, and abundant availability of water and energy. In conclusion, the use of BTF is a sustainable option for APUs despite the limiting factors identified in this research which slow down the growth of the sector. It is advisable to study Mexican producers with BFT, in order to spread their benefits to other APUs, and further evaluate the productivity of the aquaculture sector. This study considers production aspects, and also sustainable use of its resources, specifically, surface, energy, water, and food.

**Keywords:** symbiotic technologies; aquaculture sustainability; rural fish farms; Mexican fisheries; territorial diagnosis

## 1. Introduction

The General Law of Sustainable Fisheries and Aquaculture regulates fishing and aquaculture activities in Mexico. Aquaculture is defined as a set of activities aimed at controlled, pre-fattening and fattening reproduction of fauna and flora species. These activities are carried out in facilities located in fresh, marine, or brackish waters, by means of breeding or cultivation techniques, which are susceptible

to commercial, ornamental, or recreational exploitation. The law classifies aquaculture according to its objectives as follows: (a) commercial aquaculture, applied with the purpose of obtaining economic benefits; (b) promoting aquaculture for study purposes; c) didactic aquaculture for training and teaching purposes, implying people involved in aquaculture, implemented in water bodies under federal jurisdiction; (d) industrial aquaculture, as a large-scale aquatic organism production system, with a high level of business and technological development and large capital investment of public or private origin; and (e) rural aquaculture, as a small-scale aquatic organism production system, carried out by families or small rural groups, in extensive or semi-intensive crops, for self-consumption or partial sale of surplus crops [1].

At a global level, aquaculture is relevant because it is both a source of food with a high nutritional contribution for the population, and a source of economic income for millions of people [2–6]. In 2016, fish harvested from aquaculture reached up to 80 million tons, estimated at a first-sale value of USD 231.6 million. In accordance with [7], global aquaculture production was 46.8% of total production; representing almost half of the global primary production. An increasing trend in aquaculture production has been observed in all continents; China was the most important producer (that year, 2016) with 49 million tons, and the American continent, led by Chile and Brazil, had an aquaculture production of 1,562,500 tons.

Mexico, on the one hand, is one of the 25 most important countries in aquaculture production, with a production of 327,100 tons. Statistical records show a 15% growth rate, exceeding the 6% annual rate worldwide [8]. On the other hand, [9] reported that, in Mexico, there were 56,000 fish farmers who operated 9230 farms with species that contributed to guaranteeing food sovereignty such as shrimp (*Litopenaeus vanamei*), tilapia (*Oreochromis niloticus*), oyster (*Crassostrea gigas*), carp (*Cyprinus carpio*), and trout (*Oncorhynchus mykiss*). In the case of the two main species produced in Mexico, "shrimp (*Litopenaeus vanamei*) and tilapia (*Oreochromis niloticus*)", it has been possible to use biofloc technology, which was successfully applied in other studies [10–17]. There is evidence related to the main systems used in aquaculture production, i.e., extensive [6] and semi intensive [7]. These aspects are thoroughly analyzed in this study. The largest numbers of farms are tilapia, (4626); trout (1843), and shrimp (1447). The five states with the highest aquaculture production are Sinaloa, Sonora, Jalisco, Veracruz, and Chiapas with productions ranging from 62 to 30 thousand tons per year.

On a national level, the economic impact of this activity amounts to 1,644,107,167.60 USD, which represents 4% of the gross domestic product related to agriculture in 2019 [18], making it one of the main sectors for the production of food of animal origin. However, the environmental implications of this activity, especially in terms of water, have started to become an issue. First, the excessive use of water from the economic activity, given the low national availability [19,20] and, second, the pollution caused by wastewater from these aquaculture activities [21–25].

The growth of the activity considerably affects ecosystems surrounding aquaculture farms [26–30] and the environmental impacts must be minimized to ensure its sustainability, especially in Latin America, where the increase has been significant and disorderly [31–36].

According to [37,38], the water needed to produce one kilogram of tilapia (*O. mosambicus*) was between 21 and 57.7 $m^3$ of water. To avoid excessive waste, biofloc technology has been applied with good results, reducing the use of water considerably to only 1.67 and 13 $m^3$ of water to produce 1 kg of *L. vanamei* and *O. niloticus* [39,40], respectively. This reflects that the use of such technology in water management considerably minimizes the use of water resources [41]. Other studies stated, in relation to the species *O. niloticus*, that 21 $m^3$ were required and, in the case of the species *L. vanamei*, 11 to 21.43 $m^3$ were necessary under extensive conditions [42].

Regarding other species susceptible to be cultivated with BTF, no abundant information was found that estimated the water use in production. Therefore, there is a lack of scientific information about it. However, under commercial conditions, a case study was found in Veracruz, Mexico that analyzed six production units that used BTF, finding that an average of 2.07 $m^3$ of water was needed to produce one kilogram of tilapia (*O niloticus*) [43].

At the laboratory level [10], a comparison of biofloc technology (BTF) with recirculation aquaculture systems (RAS), reported that the water consumption was 1.67 m$^3$ per kilogram of produced tilapia. This information limitations may be due to the fact that other species have less economic impact. However, biofloc technology studies have been carried out on *Clarias gariepinus*, *Macrobrachium rosenbergii*, *Panaeus monodon*, *Hibird Pangasius bocourti*, and *O. aureus* [44–46].

## 1.1. Biofloc Technology: Origins

Biofloc technology was developed at the Waddell Mariculture Center (United States) in the early 1990s [47]. Subsequently, ZEAH (Zero Exchange, Aerobic, Heterotrophic Culture Systems) emerged, where [48], cited by [49,50], stated that most environmental concerns about current aquaculture practices could be effectively addressed by means of such techniques. These were implemented commercially by Belize Aquaculture Limited in the cultivation of *L. vannamei*, in Central America. In addition, the technology was originally called "active suspension ponds", defined as a new concept in water treatment. Many academics are working on research in what is now known as biofloc technology (BFT) [15,51–53].

## 1.2. Biofloc Technology: Definitions

According to [54], the development of concepts and referred applications have given rise to BFT, which has been based both on maintaining water quality conditions in relation to the fixation and controlling toxic inorganic nitrogen ($NH_4$, $NH_3$, $NO_2$, and $NO_3$). Since these nitrogenous components are particularly toxic at different levels for aquatic organisms, their disposal is very important to the development of aquatic organisms, generating on-site microbial protein that can be used as food for the cultivated species [53,55–61].

Biofloc has been described as a community made up of microorganisms associated with each other in a suspended or floating substrate (biofloc), with a density that is between 10 million and 1 billion microbial cells/cm$^3$ [54]. In this sense, [62] defined the biofloc community as having an irregular, deformable, porous shape, an indefinite size, and floating and with a tendency to settle slowly. Meanwhile, [63,64] added that each biofloc was also a micro niche with particular physiological needs, in which complementary aerobic and anaerobic processes cohabit, with interactions that were key pieces for maintaining water quality. Systems with BTF have special features that make them complex, given the amount of synergies that occur in a so-called "microcosm". Functionally, autotrophic and heterotrophic activities are developed by exotrophic contributions, and these activities are complemented by processes that either need oxygen or not. This set of relationships create unique conditions that directly influence the type of bacterial communities, their structure and diversity, among other features (Figure 1). Microbial anaerobic processes, if left unchecked, can produce compounds that are highly toxic to farmed animals [63]. Although much remains to be known, the fact that BFT treats waste conceptually, makes it a possible and environmentally friendly alternative, because at the same time that it saves water and recycles nutrients, it discharges few pollutants [50].

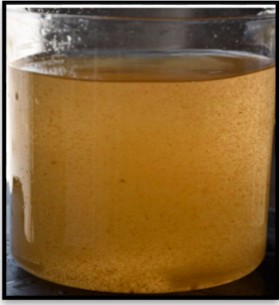

**Figure 1.** Beaker with water sample, where they observe bioflocules in suspension. Source: Photographs provided by David Celdran Sabater (2020).

Additionally, [65] (p. 21) explained that BFT focuses on the development of a community of microorganisms agglutinated in bioflocules (microorganism agglomerates formed by protozoan bacteria, phytoplankton, and zooplankton; all these forms are a biofloc) that control the degradation of toxic substances in water, and food remains, no water is exchanged, results in higher densities, improves the biosafety of the systems, provides health benefits to the organisms, saves on food consumption and water pumping, and allows the producer to use of a smaller area for cultivation. Figure 2.

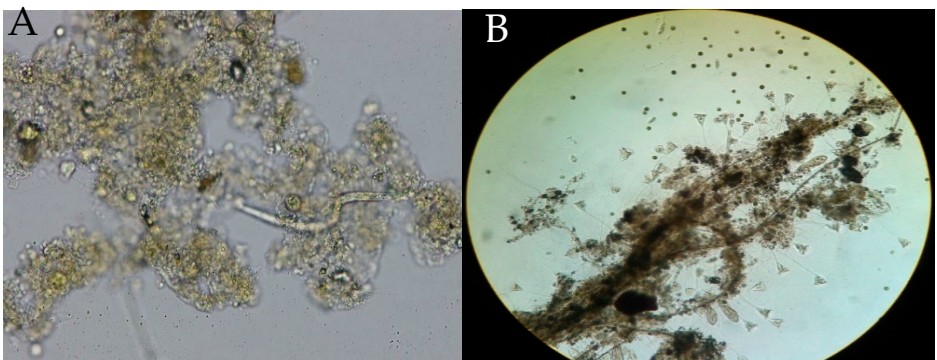

**Figure 2.** (**A**) Biofloc with *Nematoda*; (**B**) Biofloc with *Vorticella* and *Chlorella.* Source: Photographs provided by David Celdrán Sabater (2020).

The nutritional content of bioflocs includes protein in ranges from 25 to 50%, although the predominant range is between 30 and 45% [66]. Analyses have also shown that bioflocs are great suppliers of vitamins and minerals. They also have probiotic effects, and therefore are considered to be biosecure, reducing the impact on pathogenic bacteria, which are relatively low in systems with BTF [67]. The result is the presence of both cultivated organisms with less disease impact, and nitrogen-degrading bacteria such as *Nitrospira* sp., *Nitrobacter* sp. and *Bacillus* sp. In other words, these genres are associated with better maintenance of crop water quality [68].

The authors of [42,69,70] claimed that these groups of bacteria were dominant in these types of system, achieving an effective biocontrol over pathogenic microorganisms. In terms of high density, biofloc systems offer intensive production, the most frequently tested aquatic organisms such as tilapia and *L. vanamei* adapt to the conditions within the biofloc systems and thrive upon using bioflocs as a food source [71,72].

A study by [10] compared the tilapia culture Genetically Improved Farmed Tilapia (GIFT) in a biofloc system and in a recirculation system and concluded that the culture in a biofloc system was more effective than the one in a recirculation system. Studies by [71,73] agreed that tilapia biomass in biofloc systems could reach between 200 and 300 tons per hectare.

After smaller area of cultivation, advantages and disadvantages are added by the introduction of BFT. Thus, [16] and [66] highlighted that bioflocs provided essential services, such as: (a) maintained water quality by assimilating nitrogenous compounds from food waste; (b) provided a natural food source that is rich in protein, which reduces the feed conversion rate (FCR); (c) reduced food costs; (d) reduced wastewater discharges into rivers, lakes, and estuaries, and prevented the escape of animals, nutrients, organic matter, and pathogens; (e) minimizes the rate of water refilling [70]; and (f) increased production. However, [43] stated that Mexican farms faced the following disadvantages when operating with BFT: (1) Continuous use of electricity, which increased the costs for the aquaculturist. (2) Electrical power failures that were critical for cultivated organisms. (3) Geomembrane covered ponds were necessary, if not available. (4) In some cases, the construction of greenhouses was necessary to keep the temperature under control and limit seaweed growth. (5) Operation with the biofloc technology required qualified personnel. (6) Constant water quality checks with laboratory equipment were a must.

These issues are overcome with the acquired experience of operating this technology, starting with low densities to a full understand of the water quality variables that need to be controlled. It is

important to emphasize that, in regions where water availability is limited, this biofloc technology is cheaper than implementing an aquaculture recirculation system, given the construction costs [43]. This solution helps to mitigate production costs and to minimize water use in fish farms.

In this sense, the objective of this study was to identify the main factors that limit the Mexican aquaculture producer from adopting biofloc technology in their aquaculture production units (APUs).

## 2. Materials and Methods

### 2.1. Study Region: Mexico

Mexico has 9230 aquaculture production units (APUs) with the Registro Nacional de Pesca y Acuacultura (RENPA) (National Registry of Fisheries and Aquaculture), distributed throughout the federal territory. The study sample is distributed in 16 of the 31 states, covering 50% of the country (Figure 3). All the interviewed aquaculturists were dedicated to the cultivation of different species.

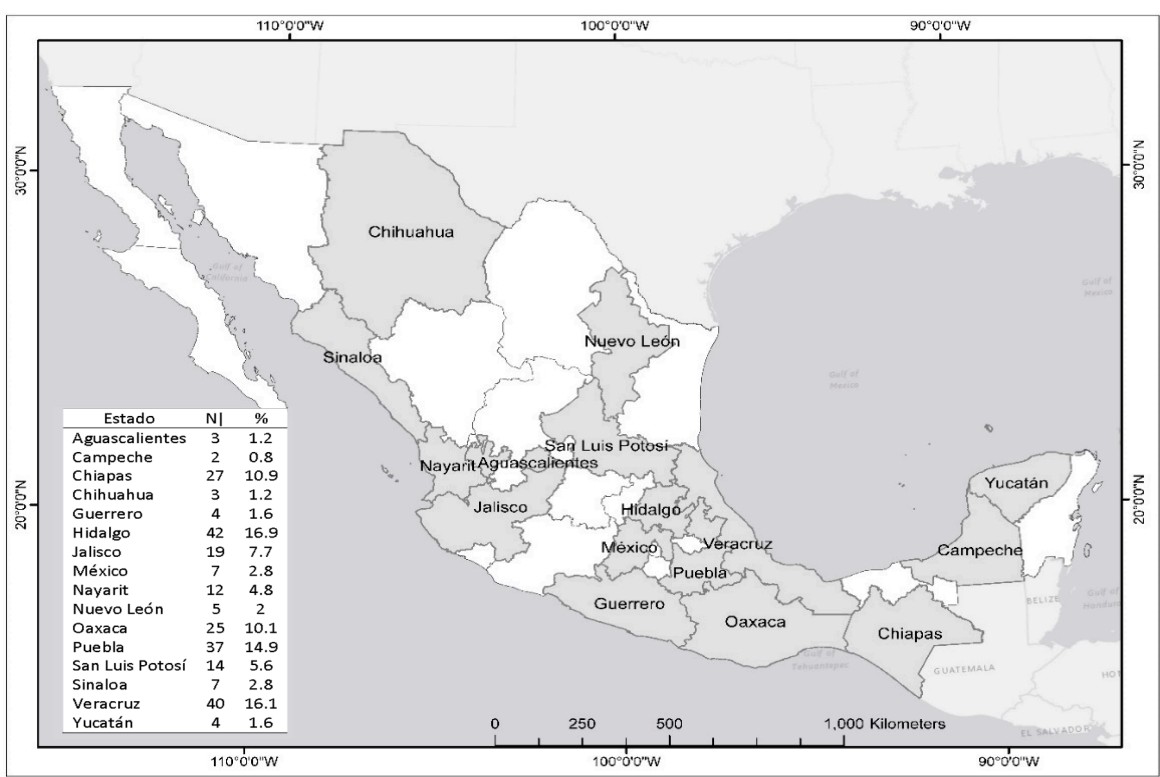

| Estado | N | % |
| --- | --- | --- |
| Aguascalientes | 3 | 1.2 |
| Campeche | 2 | 0.8 |
| Chiapas | 27 | 10.9 |
| Chihuahua | 3 | 1.2 |
| Guerrero | 4 | 1.6 |
| Hidalgo | 42 | 16.9 |
| Jalisco | 19 | 7.7 |
| México | 7 | 2.8 |
| Nayarit | 12 | 4.8 |
| Nuevo León | 5 | 2 |
| Oaxaca | 25 | 10.1 |
| Puebla | 37 | 14.9 |
| San Luis Potosí | 14 | 5.6 |
| Sinaloa | 7 | 2.8 |
| Veracruz | 40 | 16.1 |
| Yucatán | 4 | 1.6 |

**Figure 3.** Aquaculture producers surveyed in Mexico (2019).

### 2.2. Data Collection

A structured questionnaire with open and closed questions was used as a data collection instrument. Nine categories (factors or indicators) were considered, including forty-nine questions with quantitative and qualitative variables, which were subsequently coded to facilitate their analysis (Table 1). The data collection was carried out in the following three international aquaculture congresses, i.e., 2018, 2019, and the National Tilapia Forum 2019, which are the most important events in Mexico in this sector. Two hundred and forty-eight questionnaires were applied, in the three samples points, to a population of 2.63% of the total 9230 aquaculture production units (APUs), with the National Registry of Fisheries and Aquaculture.

**Table 1.** Mexican aquaculture production units registered in the National Registry of Fisheries and Aquaculture (UPA-RNPA by its initials in Spanish). Categories (2019).

| Category | Quantitative Variables | Qualitative Variables |
| --- | --- | --- |
| **General Features** | 1. Aquaculture age (EDA) | 2. Federated states surveyed (EFE) <br> 3. Level of Schooling (ESC) |
| **Socio-economic Information** | 1. Staff attached to the UPA (PAU) | 2. Alternative productive activities (APA) |
| **Production** | 1. Daily spare rate (TRD) <br> 2. Hours of water pumping (HBA) <br> 3. Production cycle (CPO) <br> 4. Weight organisms at sowing (POS) <br> 5. Sales weight (PEV) <br> 6. Lts of water per kg of product (LKP) <br> 7. Organisms per cubic meter of sowing (OMC) <br> 8. Cost of energy (CEE) <br> 9. Production cost per kilogram (CPK) | 10. Water exchange (RAD) <br> 11. Cultivated species (EPC) <br> 12. Perception of water use (PAU) <br> 13. Use of operation records (URO) <br> 14. Brand of food used (MAU) <br> 15. Physicochemical data on water quality (DFA) |
| **Normative** | | 1. National Register of Fisheries and Aquaculture (RNPA) <br> 2. Federal Taxpayers Registry (RFC) <br> 3. Water concession title (CNA) <br> 4. Water discharge permit (CNA) |
| **Technology and System Management** | 1. Experience in aquaculture (EXP) <br> 2. Diameter of discharge pipe (DTS) <br> 3. Number of ponds per unit (NEP) <br> 4. UPAS surface (SUP) | 5. Source of water supply (FAA) <br> 6. Use of discharge water (UAD) <br> 7. Type of culture ponds (TED) <br> 8. Type of aeration system (TSA) <br> 9. Electric power backup systems (SER) |
| **Commercialization** | 1. Sale price (PRV) | 2. Wholesale sales percentage (VMA) <br> 3. Retail sales percentage (VME) <br> 4. Sales system (SVE) |
| **Training** | 1. Number of congresses per year (NCA) <br> 2. Cost of technical assistance (CAT) | 3. Technical assistance received (ATR) <br> 4. Knowledge of water technology (CTE) <br> 5. Willingness to learn (DPA) <br> 6. Interest in the efficient use of water (UEA) <br> 7. Sustainability concept (CSU) <br> 8. Results of technical assistance (RAT) |
| **Organization and Suppports for Production** | | 1. Knowledge of associations (CAP) <br> 2. Government and type subsidies (SGT) |

The selection method of observation units was intentional, not random. The data collection dates were divided into the following three periods: September 2018 (105 questionnaires), October 2018 (82 questionnaires), and September 2019 (61 questionnaires). The confidence level is 90% and the margin of error is 10%.

## 3. Results

### 3.1. General Features

The data obtained on national aquaculture in 16 states of the Mexican Republic highlights the participation of Veracruz, Puebla, Oaxaca, Hidalgo, and Chiapas. These entities are among the top five in national aquaculture production.

Regarding the age of the aquaculturists (EDA), the predominant range is from 41 to 47 years with 31.5%, followed by 33 to 40 years with 25%. However, the participation of young people from 18 to 32 years old in the aquaculture activity is lower, only 9.3% (Figure 4). Consequently, there is a solid base of young people and young adults in the sector (15 to 47 years, 65.8%), where it is possible to work and change paradigms with innovative production systems. Our statement that young farmers are more likely to adopt recent technologies was in agreement with [74] and with the authors of [75], who suggested that advanced age is an obstacle to technology adoption.

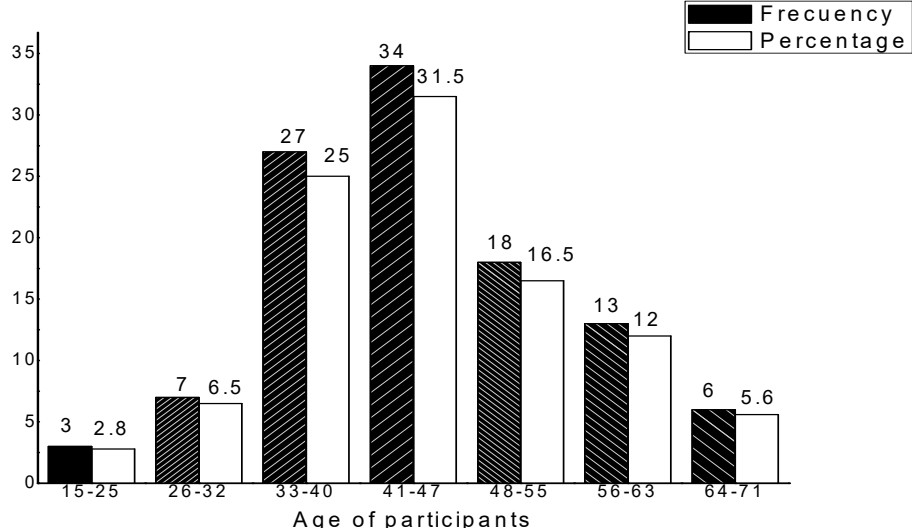

**Figure 4.** Aquaculture producers surveyed in Mexico. Age of participants (2019).

Regarding the schooling of aquaculturists (ESC), basic education stands out with 68.14%, which includes levels of no schooling, primary, and secondary. Less than 32% of the Mexican aquaculturists in the analyzed sample have intermediate and higher-level studies such as high school (18.55%), bachelor's degree (10.08%), and postgraduate (3.23%). This is a limiting factor for the technological development of national aquaculture, as there is a shortage of trained and specialized technical personnel. The low level of schooling was also recently reported by [76] in the state of Puebla and by [77] in the state of Campeche. This is a major deficiency for proper BFT performance in commercial production. Knowledge of fundamentals and their effect on cultivation is required, as pointed out by [78]. While [79] stated that age can contribute positively to the adoption of techniques, when the levels of education and experience of farmers are higher, we agree with [80] who affirmed that farmers with more and better education were willing to adopt modern technologies in the production process.

### 3.2. Socioeconomic Information

As for the personnel assigned to the production unit (APU), in Mexican aquaculture farms there are four to eight people working directly in the activity (44.4%). Furthermore, one to three people are employed on 28.6% of the farms, and there is an increasing and important employment phenomenon in the sector, as has been reported by [81].

Additionally, 62.5% of the aquaculturists carry out another alternative activity (APA) that generates income, predominantly agriculture and livestock with 28.23%, aquaponics with 14.52%, and restaurant service with 18.5%. This shows the importance of aquaculture in generating employment in the rural sector, with a tendency towards the diversification of productive activities. We agreed with [79] who pointed out that the lack of advice in different areas of activity has caused aquaculture to become a secondary activity, i.e., aquaculture in Mexico is still being developed as a complement to agricultural activities. In the case of restaurants, the aquaculturist is closing the marketing links, bringing his transformed product directly to consumers and this activity should be replicated, to strengthen the incomes of those who do not yet practice it. This approach is defended by [82–84] in Veracruz, who developed the "live tilapia concept", bringing fresh and live product to the consumer. In addition, both [85,86] concluded that aquaculture favored the diversification of economic activities; a positive aspect for the development of the Mexican rural sector that currently prevails.

*3.3. Production Information*

Water exchange (RAD), in national aquaculture, is a primary activity to maintain water quality. It is logical to find that 83.6% of surveyed farms carry out this activity, where 76.21% of them carry out their daily turnover (TRD) at least once a day. Moreover, the hours of water pumping (HBA), most frequently used to make the replacements, was two hours in 26.21% of the cases and three hours in 28.63% of the cases. It is important to point out that, in aquaculture in Mexico, there are floating cages used for production in the case of 12.10% of those surveyed, thus, making the pumped replacement unnecessary because the flow is natural. Only 4.84% recirculate all day with low water flow. These results confirm what [87,88] found in Northeast Mexico, i.e., high rates of water turnover and pumping prevail in aquaculture. This confirms that Mexican aquaculture depends on water replacements to maintain water quality, showing a strong need to implement biofloc technology in order to avoid this requirement.

Regarding production cycles (CPO) from planting to harvest, the interviewed aquaculturists cultivate different species such as *O. niloticus*, *O. mossambicus*, *L. vanamie*, *Ictalurus punctatus*, *Macrobrachium rosembergii*, *Lithobates catesbeianus*. The predominant frequencies were 29.84% with six months, 26.21% with five months and, finally, 27.82% with more than eight months. This means that more than 56.06% can perform two cycles per year. This data coincides with [89], who discovered that 71% of aquaculture companies in Nayarit carried out two annual cycles. It is important to emphasize that, with biofloc, the cycles are shortened, with two to three cycles per year for tilapia, and three cycles for shrimp. This is due to the fact that, by having continuous food (bioflocs), animals ingest more food, and therefore grow faster. In addition, biofloc crops cause animals to better digest their food and have lower feed conversion ratios (FCRs) [12,13,57,60,90].

Organism weights (POS) at planting, in the case of tilapia, are 1 g in 47.22% of cases, followed by 4 g in 21.30%. This situation differs, given that it depends on the production system. There is a nascent trend of planting pre-fattened organisms, in a low percentage (6.48%) of 10 g and 50 g (11.11%). This phenomenon, without a doubt, arises with the purpose of reducing harvest times. The sales weight (PEV) starts from 350 to 400 g for 28.70% of tilapia producers, from 400 to 500 g, for 20.37% of aquaculturists, and more than 500 g, for 26.85% of aquaculturists. This data confirms the estimates of [91], who classified the sales sizes in three categories, i.e., small, medium and large.

An important aspect in aquaculture that aims to be sustainable is the liters of water used per kilogram of fish produced (LKP). In this regard, 95% of the respondents stated that they were unaware of this information, a worrying situation since they do not keep a record of the water flow used in a production cycle. Only 5% of them answered that they used a volume that varied between 10,000 and 50,000 L per kilogram of fish.

Regarding crop productivity (OMC), in the case of tilapia, low intensity per cubic meter predominates, i.e., 39.52% of aquaculturists answered from 5 to 10, 28.63% from 11 to 20, and 24.19% from 21 to 30 organisms per cubic meter of water. In addition, only 7.66% of the aquaculturists answered that their densities were greater than 30 organisms. This shows that tilapia is the most cultivated species in Mexico and that the production systems are semi intensive in a large proportion (68.13%), with a productivity of 2.5 to 10 kg/m$^3$. This data agrees with [92] and the FAO [6,7], confirming the existence of Mexican farms operating under intensive systems with technological advances, yet a rural aquaculture is practiced with very low productivity. The authors state that this situation has not improved after 15 years and that there is a complete lack of awareness about the amount of water used to develop this activity.

However, it stood out positively that 31.85% of the systems were intensive (10 to 15 kg/m$^3$), according to the classification described by [93], who affirmed that the tilapia aquaculture in Mexico was divided into systems: extensive, from 1 to 4 fish per m$^3$ (500 g to 2 kg/m$^3$); semi-extensive from 5 to 10 fish per m$^3$ (2.5 to 5 kg/m$^3$); semi intensive from 10 to 20 fish per m$^3$ (5 to 10 kg/m$^3$); and intensive from 20 to 40 fish per m$^3$ (10 to 20 kg/m$^3$). The significance of these results was that no commercial evidence was found among respondents from super intensive farms in Mexico. In the case

of *L. Vanamei*, the predominant sowing densities of post larvae per square meter ($PL/m^2$) were extensive systems, 4–10 $PL/m^2$ (22.51%); semi-intensive systems, 10–30 $PL/m^2$ (49.30%) and intensive systems, 60–300 $PL/m^2$ (28.19%). Hyper-intensive systems that manage higher sowing densities than 300 to 450 $PL/m^2$ were not reported. This ranking was established as formulated by [94]. Although there is no consensus on the classification of farming systems in terms of productivity between [95–98] with respect to [99], the truth about national aquaculture is that they operate with low productivity as a common feature of their farming systems.

The cost of electrical energy for pumping and aeration is an important aspect in semi-intensive and intensive aquaculture systems. As such, 23.79% of the respondents answered that the cost of electric energy per month (CEE) added up to 188.45 and 235.56 USD and 17.34% answered that it goes from 282.66 and 329.77 USD. This indicates that the cost of electrical energy in Mexican aquaculture farms is high. However, since the majority of aquaculturists are registered in the National Registry of Fisheries and Aquaculture (RNPA), they have a federal subsidy of 50% of the total energy cost, through the National Commission for Fisheries and Aquaculture [100]. It is inferred that the data obtained correspond to energy costs that already include the subsidy or "energy share for aquaculture activities".

To this effect, Ref. [93] reported that the energy cost usually represented between 10% and 15% of total production costs. However, aquaculturists commented that, in Mexico, it was very expensive to pay for electricity without subsidies, which was a reason why many farms close operations. In this sense, Ref. [100] reported 458,260 GWh of energy subsidies for the sector during the fiscal year 2017, representing an amount of 8,876,087.82 USD for 504 producers in 29 states of the country. Although the Mexican government covers part of the energy costs, coverage is still limited, as the UPAS universe is larger. Energy cost is a key factor in the sum of the production costs in aquaculture, given that the price of electrical energy in commercial and industrial fees in Mexico is high [101]. One problem to evaluate and solve is that aquaculture is not considered for a special rate, as in the case of low voltage agricultural irrigation (RABT, Riego Agrícola en Baja Tensión) and medium voltage agricultural irrigation (RAMT, Riego Agrícola en Media Tensión). This aspect must be legislated so that the primary productive activity has irrigation subsidies. To exemplify, the current agricultural rate with subsidy for irrigation systems is 0.027 USD per kWh at daytime and 0.013 USD per kWh at night [101]. On the contrary, in aquaculture activity, the established rates are the small demand in low voltage (PDBT, Pequeña Demanda en Baja Tensión), large demand in low voltage (GDBT, Gran Demanda en Baja Tensión), and large demand in medium voltage (GDMTO, Gran Demanda en Media Tensión), carried out by applying charges that correspond to the rate that results correlative, multiplied by the factor of 0.50 (energy quota). That is, depending on the connected load, it can fall in rates from the PDBT, GDBT, or GDMTO, considering that most aquacultures with small farms fall within the PDBT category whose cost kWh is 0.170 USD per kWh [101]. In addition, there is no differential for day or night periods, which shows unequal treatment for such an important primary activity.

In all productive activities, it is vital to know the production cost. Thus, 58.47% of the surveyed fish farmers claimed to know the cost, and therefore they were questioned about the production cost per kilogram (CPK) exclusively in the case of tilapia; 45.5% answered that the cost fluctuated between 1.39 and 1.57 USD per kg, 32.17% from 1.12 to 1.35 USD per kg, and 22.38% from 1.57 to 1.80 USD per kg. This information reveals a difference of 0.67 USD between the lowest and highest production costs. Therefore, it is concluded that this phenomenon stems mainly from the management of the farm, coupled with the food that the species receive and their genetics. Accordingly, Ref. [102] reported 1.35 USD per kilogram as the cost of production.

In terms of cultivated species (EPC), it is relevant to note that 70.6% of respondents cultivate only one species. Tilapia (*O. niloticus* and *O. mossambicus*) production predominated with 108 aquaculturists, followed by 45 shrimp producers (*L. vanamie*), 29 rainbow trout (*Oncorhynchus mykiss*) producers, 32 channel catfish (*Ictalurus punctatus*) producers, 15 producers of ornamental fish in their different species, 17 producers of Giant freshwater prawn (*Macrobrachium rosembergii*), and only 2 bullfrog

producers (*Lithobates catesbeianus*). It is important to emphasize the existing introduction to aquaculture of other species such as *L. catesbeianus* and ornamental fish. The latter is due to a growing demand for these products in central and shallow regions where there is an Asian population that demands frog legs. Likewise, there is a growing national market for aquarium fans, which boosts the development of this sector. Meanwhile, only 23.4% of the aquaculturists have polycultured two species on at least one occasion (Figure 5).

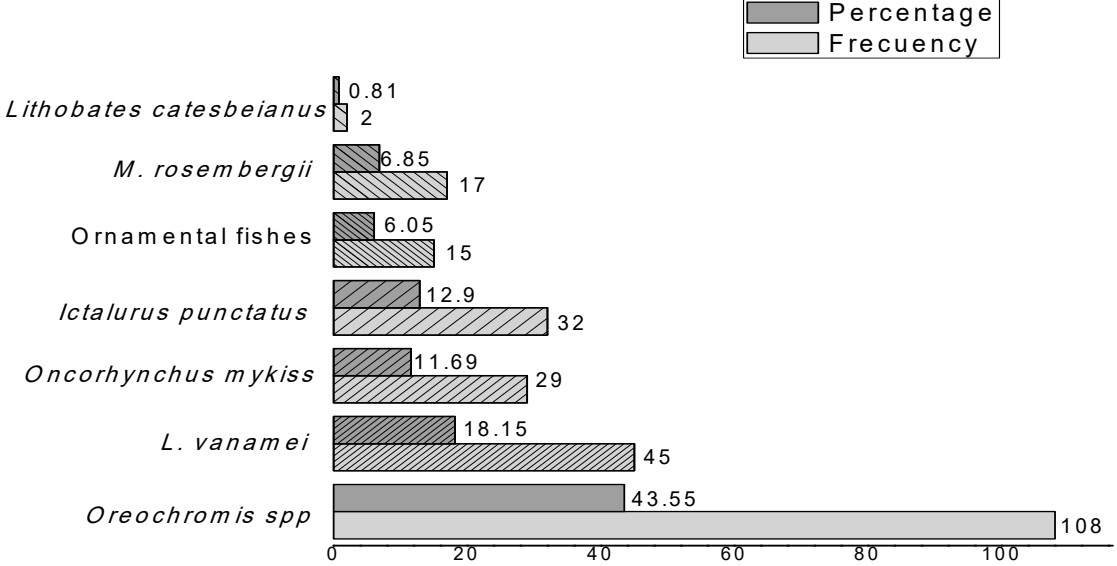

**Figure 5.** Aquaculture producers surveyed in Mexico. Species cultivated (2019).

They normally associate *L. vanamei* and *O. niloticus* and other combinations such as *O. niloticus*, *Macrobrachium rosembergii*, *O. niloticus*, and *Cherax quadricarinatus*. Tilapia has been associated with multiple species worldwide (Figure 6), as reported by Wang and Lu [103]. This polyculture has been applied, with good results, with chucumite (*Centropomus parallelus*) and snook (*Centropomus undecimalis*), according to [104], with red tilapia (*Oreochromis spp.*), and red carp (*Ciprynus carpio*) according to [105], and acocil or river crab (*Procambarus acanthophorus*) with Nile tilapia (*Oreochromis niloticus*), as indicated by [106].

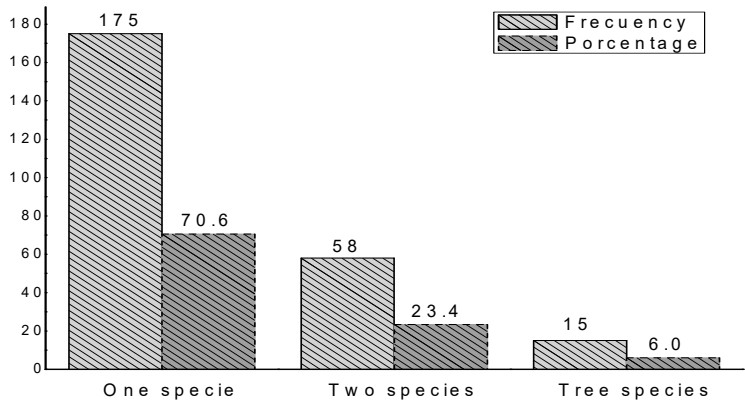

**Figure 6.** Aquaculture producers surveyed in Mexico. Polyculture practice of aquatic species (2019).

A very important aspect for the development of aquaculture is the use of water. In this sense, 34.4% of aquaculturists in Mexico perceive the use of water (PUA) as regularly excessive, 22.98% as little excessive, and 25% affirm that it is not excessive. These statements show that efficient water use is not a priority. Such insight comes mainly from the lack of information and insufficient water use records, where the large volume of vital liquid necessary for daily operations on an aquaculture farm

could be seen. Only 17.34% of the respondents consider that the use of water in aquaculture is very excessive, mainly in the northern states of the country. This is consistent with [107], who assert that, in Mexico, of every 100 L of water used, 76 L were used in agricultural activities. This primary sector included agricultural irrigation and aquaculture, where the activity with the highest discharge volume was aquaculture.

This situation is worrisome and is reflected in the use of operation records (URO). Our results show that 44.34% of the aquaculturists do not keep any type of record that allows them to numerically observe the farm's behavior, such as growth curves, feed conversion factors, body condition factors, kilograms produced per unit area, costs and profits, among others.

However, there is a daily discipline in obtaining physicochemical data from water (DFA). It was observed that 62.5% of the respondents collect data, the verified variables being oxygen (37.42%), temperature (23.23%), pH (14.34%), and ammonium (10.97%). The methods to carry out these analyses, in 62.50% of the cases, consist of digital portable equipment. Aquaculturists are undoubtedly aware of the need for water quality control for each species.

Aquaculture also depends on balanced foods, having a wide range of brands that offer this service in the country. According to the respondents, the three most frequently used food brands (MAU) are Pedregal (27.02%), Purina (20.97%), and Winfish (20.16%). Therefore, there is plenty of available fish food in the country, becoming a crucial input in aquaculture production, since its use represents between 50 and 70% of production costs, as indicated by [108] and [109]. However, it is necessary to evaluate both the quality of these foods and their effects during the development of the crop.

### 3.4. Regulatory Aspects

In every productive activity, it is very important to comply with the applicable regulations of Mexican aquaculture producers. Although [74] had 9230 producers registered (official aquaculture), in this study, the sample indicated that only 89.10% had the Federal Taxpayer Registry (RFC). This indicates that nine out ten producers are within the formal economy and belong to the primary gross domestic product generated in the country. Furthermore, the National Water Commission regulates the use of water in Mexico [10]. In this study, 75.40% of the surveyed aquaculture producers had the title of water concession for aquaculture use (CNA), 19.35% had their title pending, and the rest (5.24%) did not have it. In this sense, compliance by the majority of aquaculturists is confirmed.

However, in the case of wastewater discharges from aquaculture (CNA), only 32.26% had a permit, 20.56% affirmed that it is in process, and 47.18% did not have it (Figure 7).

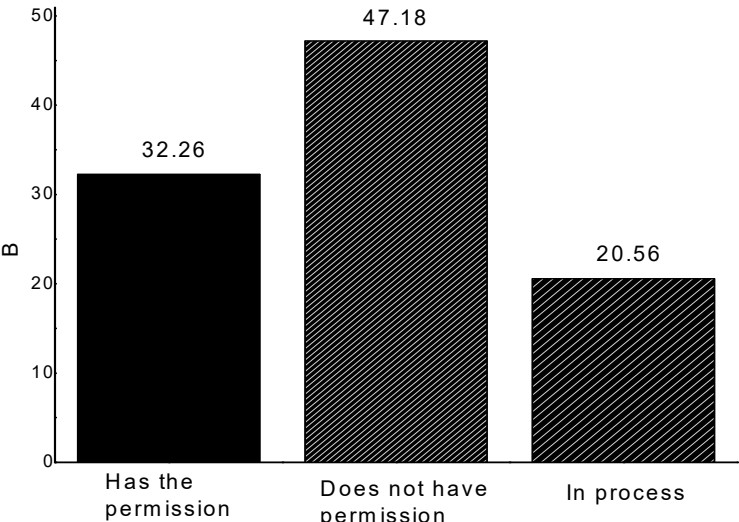

**Figure 7.** Aquaculture producers surveyed in Mexico. Percentage of wastewater discharge authorizations (2019).

These findings reveal some flexibility from the water authorities, i.e., seven out of ten producers do not have a permit for wastewater discharges, considerably affecting the public tributaries where they are discharged. This situation occurs mainly due to the fact that this download permission is closely related to the size of the farm. That is, small farms, according to the legislation, do not need to have an Environmental Impact Manifesto, since, due to their location, dimensions, characteristics, or scope, they do not produce significant environmental impacts. In other words, small farms are not required to treat wastewater discharges. In this regard, we agreed with [92], who stated that the regulations for the aquaculture sector have not been clear and have generated structural deficiencies, as well as organizational and technological limitations in assistance and training.

According to [110], this failure to comply with current regulations could be attributed to three main reasons: (1) political flexibility; (2) marginalization and poverty in areas where the activity is carried out; and finally, (3) the belief that the activity, being small, "does not pollute". These facts revealed important data obtained by the transparency portal of [19], which stated that there are 734 wastewater discharge concessions in the country for aquaculture use. On the one hand, if we compare this with the sum of 9230 registered aquaculturists, we can affirm that only 12.6% of the farms are regulated, that is, 87.4% of the aquaculturists (practically nine out of ten) do not comply with the regulations concerning wastewater discharge. On the other hand, there are officially 1506 concession titles (16.3%) for groundwater and surface water for aquaculture use in the country. This indicates that 7724 aquaculture production units (83.7%) probably do not have them or are working with agricultural concessions.

## 3.5. Technological Information and System Management

It is important to identify the resources that aquaculturists have to migrate to biofloc technology (Table 2). On the basis of the results of this section, important strengths have been identified, the first of which focused mainly on the experience of aquaculturists in the activity (EXP). This aspect is vital, since the familiarity and empirical knowledge acquired by the most experienced is essential. We also noticed that there is a similar proportion that starts in the activity, in other words, there are two important groups in the Mexican UPAS, i.e., the new aquaculturists (1 year) and those with extensive experience (more than 5 years). This allows us to affirm that there is a growing interest in the activity in the country and that it is important to promote the experiences and knowledge of these entrepreneurs.

This new generation of aquaculturists requires learning new production technologies with a sustainable approach. It is important to accumulate knowledge and experience with farming systems obtained from years of observing and experiencing how older farmers work, as [111] pointed out. However, Ref. [112] classified the older and more experienced groups as "technological laggards", because they tended to view the new technologies with skepticism. We believe it is essential to attend to fish farmers who are starting out, as [75] indicated, because the ability to acquire and process complex information, education, technical knowledge and experience, also influences the decision to adopt new technologies.

In order to know the amount of water used in an APU, according to the diameter of the discharge pipe in the pumping (DTD), three main diameters were identified. Aquaculturists pump from 4 L per second (2"), 16 L per second (4"), and 36 L per second (6"). This data, when related to the frequency of the replacements, allows us to estimate the amount of water used in a day for aquaculture activity (Table 3).

**Table 2.** Main results of the Mexican UPA. Technological category and system management (2019).

| Quantitative | | | | Qualitative | | |
|---|---|---|---|---|---|---|
| (1) EXP | 30.24%<br>33.06% | 1 year<br>More than 5 years | (5) FAA | 39.52%<br>17.34%<br>12.90% | Deep well<br>River<br>Waterhole (Noria) |
| (2) DTD | 40.37%<br>23.85%<br>13.76% | 4 inches<br>2 inches<br>6 inches | (6) UAD | 52.02%<br>22.18%<br>15.73% | No use<br>Agricultural irrigation<br>It treats and reuses |
| (3) NEP | 33.87%<br>28.23% | 6 to 11 ponds<br>1 to 5 ponds | (7) TEC | 40.73%<br>23.39%<br>12.10% | Circular membranes<br>Rustic land<br>Floating cages |
| (4) SUP | 47.98%<br>33.47% | Less than 1 ha<br>1 to 5 ha | (8) TSA | 41.48%<br>27.41%<br>21.48% | Regenerative blower<br>Pallets<br>$O_2$ Air |
| | | | (9) SER | 75.56% | Yes |

(1) EXP, experience in aquaculture; (2) DTD, diameter of the discharge pipe; (3) NEP, number of ponds; (4) SUP, surface of production units; (5) FAA, water source; (6) UAD, use of discharge water; (7) TEC, type of culture pond; (8) TSA, type of aeration system; (9) SER, electric power backup system.

**Table 3.** Estimate of water use, according to discharge diameter and replacement rate.

| Spare Parts Frequency Predominant (76.21%) | Predominant Pumping Hours | Predominant Pipe Diameter | Maximum Flow According to Discharge Diameter | Daily Flow of Water Use of a Mexican UPA | Annual Water Flow of a Mexican UPA |
|---|---|---|---|---|---|
| Once a day | 2 h (26.21%) | 2″ (23.85%) | 4 L per second | 28,800 L per day | 10,512 m$^3$ |
| Once a day | 3 h (28.36%) | 4″ (40.37%) | 16 L per second | 172,000 L per day | 62,780 m$^3$ |
| Once a day | 4 h (19.35%) | 6″ (13.76%) | 36 L per second | 518,400 L per day | 189,216 m$^3$ |

The percentages do not add up to 100% because they took the most representative ones.

The surveyed aquaculturists use high volumes of water for their productive activities, ranging from 10,512 to 189,219 m$^3$ per year. This shows that it is crucial to integrate new technologies for an efficient use of the vital liquid. However, with data from CONAGUA [11], it was found that the 734 existing permits for aquaculture farms dischargeed 20,148,950 m$^3$ daily. Regarding the number of ponds per production unit (NEP), they were mostly small production units, where 62% had from 1 to 11 ponds. This indicates that the majority of Mexican aquaculturists are small aquaculture producers; the remaining 38% corresponds to medium and large producers, whose production units range from 12 to 22 cultivation ponds.

With regard to the surface of production units (SUP), it showed that the majority were surfaces smaller than 1 ha, which was consistent with the variable (NEP). This confirms that there are indeed small APUs that predominate, given the cultivated area and number of production ponds available. This situation was also reported in the Western Region of Mexico by [113], where there were small farms with insufficient financial and material resources.

As for water sources used by the APUs (FAA), most of them used reliable sources that guaranteed the quality of water required for aquaculture, as indicated by [114,115].

The water supply for the APUs varies, i.e., 53% use groundwater (deep well, waterwheel) and 46% use surface water (river, stream or runoff, reservoir, irrigation district). Only 1% use the public network. According to [19], there are 1535 aquaculture concessions in the country with a volume of 1,157,672,051.86 m$^3$ that includes underground and surface applications. This is beneficial for the activity, since the groundwater offers better quality and reliability, in addition to the fact that its temperature remains constant throughout the year and serves as a means of control to prevent the entry of pathogens or contaminants into the ponds.

Regarding the use of discharge water (UAD), findings reveal that after staying in the culture pond, the water is discarded (Table 4). It highlights that more than 50% of the cases do not have any use for it. Some aquaculturists claim to use such water for agricultural irrigation and only a small percentage (15.73%) carry out actions to treat and reincorporate the water into the aquaculture or agricultural system. Likewise, the amount of nutrient discharge that the Mexican APUs dispose of was estimated, in order to raise awareness of the importance of their treatment and the use of BFT or other technologies that help mitigate the environmental effects of aquaculture.

**Table 4.** Main results of Mexican aquaculture production units (APUs). Estimation of nutrient discharges, according to discharge diameter and replacement rate.

| Annual Discharge of Water from a Mexican UPA | Nutrient Content | Organic Material | | Total Nutrients Downloaded per Year |
|---|---|---|---|---|
| | TAN (mg/L) | DQO (mg/L) | DBO (mg/L) | TAN (kg) |
| 10,512 m$^3$ | 18.3 * | 2760 * | 5510 * | 192.36 |
| 62,780 m$^3$ | 18.3 * | 2760 * | 5510 * | 1148 |
| 189,216 m$^3$ | 18.3 * | 2760 * | 5510 * | 3462 |

Source: Own elaboration with data obtained from [97,116].

The total content of ammonia nitrogen (TAN), biochemical oxygen demand (BOD), and chemical oxygen demand (COD), commonly discharged from aquaculture waters to receiving bodies, derives from the low efficiency that aquaculture species have when converting food protein into meat. This information coincides with [12] who reported that 60% of the nitrogen contained in the food was disposed of and transferred directly to the receiving bodies.

Additionally, culture ponds (TEC) mainly handle circular, rustic earth membranes and floating cages. In small proportions, there are also rectangular and circular concrete cages. This shows a great preference in Mexican production systems for circular ponds with membranes, which are the most common due to their price, durability, and installation flexibility.

In relation to the types of aeration systems (TSA) used in the Mexican aquaculture industry, the regenerative blowers, paddle aerators, and air injectors stand out, each with different operating features and efficiency in oxygen transfer. Blowers can partially oxygenate different ponds with a single equipment; therefore, these are used in small Mexican farms.

Finally, an electrical backup system (SER) is very important for the safe operation of the aeration systems, a positive and widespread practice in the surveyed Mexican APUs, since a failure in the supply of electrical power in a semi-intensive system is deadly to fish and, especially, in a system with biofloc technology.

*3.6. Marketing*

In relation to marketing, tilapia producers, being the largest number of respondents, stand out in the price differential between retail and wholesale. This indicates that retail sales predominate, motivated by a low production scale and the best price this represents (Table 5).

At a national aquaculture level, this data matches the results presented above, i.e., retail sales have a greater representation. Regardless of the scale of production, sales' systems are diversified in an unbalanced way, i.e., they focus on farm-based sales. This may appear as a weakness, yet it is also a strength, since consumers prefer fresh products and buy them directly from UPA facilities. In contrast, Ref. [89] discovered that 65% of aquaculture companies in Nayarit sold only to wholesalers, while the remaining 35% sold some fraction of their production at retail, at the farm level. In addition, in Veracruz, Ref. [116] reported that 55% of product sales were at the farm, while 10% were local, 21% were sold in the municipal seat, 10% around the state, and 4% in the country's capital.

**Table 5.** Main results of the Mexican UPAs. Marketing category.

| Category | Quantitative | | | Qualitative | |
|---|---|---|---|---|---|
| (1) PRV * | 29.63% | Wholesale from 1.62 to 2.02 USD | (2) VMA * | 26.61% | Wholesale |
| | 70.37% | Retail 2.52 USD | (3) VME * | 73.39% | Retail |
| | | | (4) SVE * | 48.80% 21.37% 9.27% | Farm Foot Local market Dealers |

PRV, sale price; VMA, percentage of wholesale sales; VEM, percentage of retail sales; SVE, sales' system. * Tilapia producers' data.

An important and slightly forgotten aspect lies in the production of healthy foods. Tilapia grown in biofloc is healthier than the one grown in traditional systems, since bactericides, medicines, and fertilizers are not added to the water, and therefore it is considered to be an ecological product. This is due to the fact that biofloc technology (BFT) transforms waste substances (nitrates, nitrites, ammonium, and organic matter) from the aquatic production system into natural food, as stated by [71,72]. Sensory studies are needed to determine the flavors of tilapia cultured with biofloc technology and to certify these productions as organic products, as indicated by [117]. This may open markets, which are increasingly in demand in Mexico, with the USA being the main purchasing partner for tilapia. This is unknown to aquaculturists, although the market potential and the possible price premium associated with this type of product is more than attractive, i.e., investing in water resource conservation and health implies profits for the producer.

*3.7. Training, Organization, and Production Support*

In terms of training, Mexican fish farmers take part in congresses (NCA, number of congresses per year). All of them state that they attend this type of event at least once a year, since it allows them to meet other producers and share experiences and knowledge. The aquaculturists declared having received and paid technical assistance (ATR and CAT), where the cost ranged from 10,000 to 30,000.00 pesos per month. This high cost range is related to the production volume of each UPA (Table 6).

In this sense, regarding knowledge of water management technology (CTE), the three best known techniques by the respondents are aquaculture recirculation systems, aquaponic systems, and biofloc systems. This confirms the sector's tendency to be informed on aquaculture alternatives with sustainable water management principles.

Another very important aspect among aquaculturists is the willingness to learn (DPA); among the aquaculturists, 90% expressed interest in learning about these alternatives to implement them on their farms for the following three reasons: saving energy in pumping, saving water for their system, and increasing their production. The highlighted aspect is the efficient use of water (UEA). However, some aquaculturists are wary of using these emerging technologies, mainly due to technical assistance outcomes (RAT), i.e., lack of commitment, experience and knowledge on the part of the professional service provider (PSP).

Given the complexity of implementing BFT, other alternatives are emerging, biofloc versions that use ferments (aquamimicry). Thai people have been doing this for years with very good results, according to [117].

In relation to organization, most of the producers of this union declare to know some producer association (CAP). These include state aquaculture health systems, which are in contact with the union, dealing with issues that are related to health and safety of cultured and commercialized organisms.

Lastly, in terms of production supports, the government subsidies and their type (SGT), which 56% of the surveyed aquaculturists have received, have their origin in the federal and state government in

four areas, i.e., electrical power, infrastructure, genetic material and balanced food. Local governments do not participate in the development of the sector.

**Table 6.** Training, organization, and production support category.

| Category | Quantitative | | | Qualitative | |
|---|---|---|---|---|---|
| Training | (1) NCA (2) CAT | 30.24% 25% 22.58% —— 45.28% 28.30% | 1 event 3 events 4 events 899.80 to 1,349.71 —— 449.90 to 854.81 | (2) ATR * (3) CTE (4) DPA (5) UEA (6) CSU (7) RAT | 57.26% Yes<br>37.10% RAS<br>30.65% Aquaponics<br>22.58% Biofloc<br>93.55% Yes<br>92.24% Yes<br>78.63% Yes<br>31.28% Be ecological<br>29.23% Save natural resources<br>—— ——<br>26.06% Did not work<br>35.14% Not committed<br>24.32% He had no experience<br>18.92% They did not understand |
| Organization | | | (1) CAP | | 70.97% Yes<br>47.77% Aquaculture health<br>34.06% Product system |
| Production Supports | | | (2) SGT | | 56.45% Yes<br>——<br>47.86% Electric power<br>30.00% Infrastructure<br>12.86% Food<br>9.29% Genetic material<br>——<br>40.11% State support<br>59.89% Federal support<br>0% Municipal support |

QUANTITATIVE: (1) NCA, Number of conferences per year; (2) CAT, Cost of technical assistance US dollars (CAT *: Receive technical assistance); QUALITATIVE: (2) ATR, Technical Assistance Received; (3) CTE, Knowledge of water management technology; (4) DPA, Willingness to learn; (5) UEA, Interests in the efficient use of water; (6) CSU, Concept of sustainability; (7) RAT, Results of technical assistance; (1) CAP, Knowledge of producer associations; (2) SGT, Government Subsidies and their type.

## 4. Conclusions

Although the use of biofloc technology (BFT) is a sustainable option for Mexican UPAs, we identified several obstacles that limit its adoption, such as schooling and being an alternative economic activity, which is the case for 62% of the farms studied. Investing in biofloc Technology does not seem to be a priority for them.

As for production factors, since most of them do not know their production cost and the volume of water used, they have no evidence of the possible benefits that this technology can provide. The study also reveals that the perception of water use can influence the adoption of BTF. As the regulatory aspects are flexible in Mexico, they do not help to promote techniques for the efficient management of water and its discharges. In addition, the low scale of production and good retail prices limit the risk of increasing the volume of production.

Most fish farmers have little access to capitalization, mainly due to the small sizes of their farms and their sales' volumes. Finally, an open complaint is made to the technical assistance received. There are no specialized BFT extensionists in Mexico.

In contrast, some strengths in the sector stand out. Factors that could favor adoption were identified, including the age of aquaculturists (18 to 40 years old), likely motivated to enter this new form of production and interested in alternative technologies. As for production factors, the high rate

of water replacement prevails, making it feasible to apply BFT to mitigate this practice. Similarly, low productivity can be increased to triple production thanks to BFT. The electric energy subsidy is an important asset to implement the technology, essential to minimize costs. Likewise, the widespread practice of aquaculturists to take physicochemical data from water is a positive factor when migrating to BFT. Regarding regulatory factors, since applicable legislation is not complied with due to irregular wastewater discharges from aquaculture, it may be an incentive to adopt the technology to avoid possible sanctions. Another positive aspect for developing aquaculture with BFT is the high-quality water supply (reliable) and the interest in training.

The identified negative factors have solutions. These range from political to economic and social and are related to small and medium producers. The evidence presented in this study shows that it is possible to apply biofloc technology under commercial conditions. It is crucial to bring the small and medium producers who use this technology closer to the producers who use the replacement as a means of improving water quality; this approach will most likely encourage the adoption of BFT. The aquaculture producer can corroborate that BFT works and is replicable among a larger number of producers, especially *O. niloticus* and *L. vanamei* producers, which are the predominant ones.

Additionally, the academy has an alternative to transfer and promote the use of this technology that contributes to regional sustainable development, promoting aquaculture with a responsible use of water.

Until now, financing for high-tech aquaculture projects in Mexico has been limited by the high risk that the activity represents. However, with the current infrastructure available in Mexico, investments should focus on aeration equipment, backup alternative energy (solar), and laboratories that monitor water quality. Access to credit should also be facilitated, which would be quick to approve and repay in the short term due to the increase in the production levels that would be obtained, given the high demand for aquaculture products in Mexico. The time to promote BFT in the country is now.

As a recommendation, we suggest studying Mexican producers with BFT and their contributions to the sustainability of aquaculture, in order to spread its benefits for the sector. We also recommend evaluating the productivity of the aquaculture sector based on the use of resources (surface, energy, water, and feed), rather than just production aspects as is traditionally the case.

**Author Contributions:** Conceptualization, E.A.B.-T., M.d.l.Á.P.-Á., R.H.G.-M., and I.R.-M.; methodology, E.A.B.-T., and M.d.l.Á.P.-Á.; validation, E.A.B.-T., L.C.S.-H., M.d.l.Á.P.-Á., and I.R.-M.; M.d.l.Á.P.-Á.; formal analysis, E.A.B.-T., M.d.l.Á.P.-Á., and L.C.S.-H.; investigation, E.A.B.-T., M.d.l.Á.P.-Á., L.C.S.-H. and I.R.-M.; A.M.-N.; resources, E.A.B.-T., M.d.l.Á.P.-Á., L.C.S.-H., and A.M.-N.; data curation, E.A.B.-T., L.C.S.-H. and M.d.l.Á.P.-Á.; writing—original draft preparation, E.A.B.-T., M.d.l.Á.P.-Á., L.C.S.-H., R.H.G.-M., and A.M.-N.; writing—review and editing, E.A.B.-T., M.d.l.Á.P.-Á., and L.C.S.-H.; visualization, E.A.B.-T., M.d.l.Á.P.-Á., L.C.S.-H., R.H.G.-M., and I.R.-M.; supervision, E.A.B.-T., M.d.l.Á.P.-Á., L.C.S.-H., A.M.-N., and I.R.-M., project administration, E.A.B.-T., M.d.l.Á.P.-Á., and L.C.S-H.; funding acquisition, E.A.B.-T., and M.d.l.Á.P.-Á. All authors have read and agreed to the published version of the manuscript.

**Funding:** The study "Aquaculture in Mexico and the use of biofloc technology as a sustainable alternative: analysis of adoption, development and comparison with other technologies for tilapia (Oreochromis niloticus). PhD thesis in Regional Sustainable Development. Xalapa, Veracruz, Mexico: El Colegio de Veracruz." received external funding from the Mexican Consejo Nacional de Ciencia y Tecnología, Programa Nacional de Posgrados de Calidad (CONACYT, National Council for Science and Technology).

**Acknowledgments:** Thanks to Eng. Saúl Antonio Rivera González, for his support for sampling activities along the project. Jose Luis Marín Muñiz, for his collaboration in structuring the survey questions and validating the instrument, and David Celdrán Sabater for his collaboration in locating aquaculture companies that currently apply biofloc technology in Mexico and the valuable comments to the manuscript about biofloc technology.

**Conflicts of Interest:** The authors declare no conflict of interest.

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
