# Peer review of "Factors That Limit the Adoption of Biofloc Technology in Aquaculture Production in Mexico"

_water, doi:10.3390/w12102775_

Round 1

Reviewer 1 Report

Dear authors,

First of all, I would like to congratulate you. Your research details the factors that can limit and promote the implementation of BFT in Mexico.

In my opinion, some factors should be detailed and clrify if you are analysing tilapa, shrimp or other farms. It's possible that conclusions are different depending on each species. Also, yous should break down the results and discussion to make easier to read. A good option may be the use of more tables and graphs to summarize the information.

Finally, I envourage you to improve your manuscript and accept this comments and notes (detailed in the manuscript). 

Buena suerte en su investigación.

Author Response

Reviewer 1. Replies to reviewers' comments in text

Dear reviewer, we thank you for improve the manuscript. We take very carefully the comments and notes detailed in the lines of the manuscript (Manuscript.v2)

Reviewer 2 Report

The manuscript has been written well.

In my opinion few minor corrections are necessary.

In introduction, it is better to mention about the present state of aquaculture in Maxico and why BFT is required.

Advantages of biofloc should be mentioned.

Authors should emphasize on solution for BFT.

Author Response

Reviewer 2. I recommend adding advantages and disadvantages of biofloc technology.

Line 108. After smaller area of cultivation, advantages and disadvantages of Biofloc technology are added in the introduction. Thus, (44) and Hargreaves (2013) highlight that bioflocs provide essential services, such as:  

1.-Maintaining water quality by assimilating nitrogenous compounds from food waste.

2.- Providing a natural food source that is rich in protein, which reduces the feed conversion rate (FCR);

3.- Reduced food costs.

4.- The reduction of wastewater discharges into rivers, lakes and estuaries, preventing the escape of animals, nutrients, organic matter and pathogens.

5.- Minimizing the rate of water refilling (Crab et al., 2012)

6.- Increased production.

However, Betanzo Torres et al 2019 state that Mexican farms face disadvantages when operating with the BFT:

1.- Continuous use of electricity, which increases the costs for the aquaculturist;

2.- Electrical power failures are critical for cultivated organisms.

3.- Geomembrane covered ponds are necessary, if not available.

4.- In some cases, the construction of greenhouses is necessary to keep the temperature under control and limit seaweed growth.

5.- The operation with Biofloc technology requires qualified personnel.

6.- Constant water quality checks with laboratory equipment are a must.

These issues are overcome with the acquired experience of operating this technology, starting with low densities to fully understand the water quality variables that need to be controlled.

It is important to emphasize that, in regions where water availability is limited, this biofloc technology is cheaper than implementing an aquaculture recirculation system, given the construction costs (Betanzo-Torres, 2019). This solution helps mitigate production costs and minimize water use in fish farms.

New reference:

Betanzo-Torres, Erick Arturo (2019). Aquaculture in Mexico and the use of biofloc technology as a sustainable alternative: analysis of adoption, development and comparison with other technologies for tilapia (Oreochromis niloticus) culture. PhD thesis in Regional Sustainable Development. Xalapa, Veracruz, Mexico: El Colegio de Veracruz.

Round 2

Reviewer 1 Report

Thanks for accept some of my recomendations